# Metabolome-Wide, Phylogenetically Controlled Comparison Indicates Higher Phenolic Diversity in Tropical Tree Species

**DOI:** 10.3390/plants10030554

**Published:** 2021-03-16

**Authors:** Guille Peguero, Albert Gargallo-Garriga, Joan Maspons, Karel Klem, Otmar Urban, Jordi Sardans, Josep Peñuelas

**Affiliations:** 1CSIC, Global Ecology Unit CREAF-CSIC-UAB, E-08913 Bellaterra, Spain; albert.gargallo@gmail.com (A.G.-G.); j.maspons@creaf.uab.cat (J.M.); j.sardans@creaf.uab.cat (J.S.); josep.penuelas@uab.cat (J.P.); 2Ecological and Forestry Applications Research Centre, Autonomous University of Barcelona, E-08913 Cerdanyola del Vallès, Spain; 3Departament de Biologia Animal, Biologia Vegetal i Ecologia, Universitat Autònoma de Barcelona, E-08193 Bellaterra, Spain; 4Global Change Research Institute, Czech Academy of Sciences, CZ-60300 Brno, Czech Republic; klem.k@czechglobe.cz (K.K.); urban.o@czechglobe.cz (O.U.)

**Keywords:** antagonistic interactions, bayesian phylogenetic models, latitudinal biodiversity gradient, metabolomics, phenolics, plant defense

## Abstract

Tropical plants are expected to have a higher variety of defensive traits, such as a more diverse array of secondary metabolic compounds in response to greater pressures of antagonistic interactions, than their temperate counterparts. We test this hypothesis using advanced metabolomics linked to a novel stoichiometric compound classification to analyze the complete foliar metabolomes of four tropical and four temperate tree species, which were selected so that each subset contained the same amount of phylogenetic diversity and evenness. We then built Bayesian phylogenetic multilevel models to test for tropical–temperate differences in metabolite diversity for the entire metabolome and for four major families of secondary compounds. We found strong evidence supporting that the leaves of tropical tree species have a higher phenolic diversity. The functionally closer group of polyphenolics also showed moderate evidence of higher diversity in tropical species, but there were no differences either for the entire metabolome or for the other major families of compounds analyzed. This supports the interpretation that this tropical–temperate contrast must be related to the functional role of phenolics and polyphenolics.

## 1. Introduction

Attempting to explain the latitudinal gradient of plant diversity has been a major focus of interest since the age of the pioneer naturalists [1]. Building upon these early observations, the modern synthesizers laid out the hypothesis that the strength of biotic interactions, particularly antagonistic relationships with competitors, herbivores and pathogens, increase toward the tropics [2]. Accordingly, these stronger biotic interactions in the tropics must have had consequences in diversity generation accelerating diversification through coevolution and arms races [2,3,4]. However, they also must have consequences in diversity maintenance increasing the species-carrying capacity of tropical ecosystems through multiple mechanisms that facilitate species coexistence, such as a better niche partitioning or increased niche packing and stronger negative density-dependent interactions [3,4,5,6,7]. However, all these arguments are sustained by a simple though sometimes contentious prediction, i.e., tropical species must be better defended against antagonistic agents because these are more diverse and exert a greater pressure [2]. In the case of tropical plants, the wide array of secondary metabolic compounds is perhaps the most paradigmatic example of traits against herbivores and pathogens [8,9].

Many aspects of this overarching hypothesis have been well supported so far [10,11,12,13,14,15,16,17], but recent global studies and meta-analyses have questioned the evidence for some of its main predictions, for instance claiming that tropical plants are neither better defended against nor more damaged by herbivores [18,19]. Some suggestions have been posed to reconcile these conflicting views [20]. Accounting for the entire complexity of plant metabolomes, instead of focusing only on specific chemical compounds, could provide a broader perspective when conducting tropical–temperate comparisons [21]. Additionally, carefully considering the phylogenetic relationships between the species included in the analysis by properly applying comparative methods should allow to assess plant defenses within an evolutionary context [22].

Secondary metabolism in plants produces thousands of compounds in nearly 20 broad chemical families [23]. This extensive phytochemical diversity has been both the main driver of interest and the main hindrance to the comprehensive study of plant defense. The advances in mass spectrometry and nuclear magnetic resonance allow for unprecedented characterizations of complete metabolomes, which are renewing interest in the links between taxonomic and phytochemical diversity [21,24,25,26]. In addition to these breakthroughs in analytical chemistry, the development of Bayesian comparative methods facilitates the modeling of the phylogenetic structure of the data and the inclusion of intraspecific variability in trait analyses, thereby seamlessly incorporating the evolutionary relationships among species within the inferential process [27].

Here, we test the hypothesis that tropical tree species present a higher diversity of secondary metabolites in their leaves using state-of-the-art liquid chromatography and mass spectrometry (LC–MS) linked to a novel, stoichiometrically based classification of compounds into the major chemical families [28]. We selected eight tree species from distinct tropical and temperate distributions while controlling for the phylogenetic diversity and evenness within each subset of species, and by means of Bayesian phylogenetic multilevel models we tested the prediction that the richness and diversity of metabolites should be greater for species at lower latitudes, particularly for the families of chemical compounds associated with protection against antagonistic biotic interactions, such as phenolics and polyphenolics. Additionally, we analyzed the similarity of the complete foliar metabolomes and of their phenolic profiles to check for differences according to region of origin and to assess variations within and between species in chemical composition.

## 2. Material and Methods

### 2.1. Species Selection and Sample Processing

To exclude any evolutionary effect on the metabolomic differences between tropical and temperate species (e.g., contrasting species ages or patterns of diversification leading to a recent escalation or divergence of defenses [29]), we selected four tropical and four temperate tree species so that each subset encompassed the same amount of phylogenetic diversity and evenness, i.e., the total branch length and mean distances between the species in each phylogenetic subtree were equivalent (Table 1 and Appendix A). We collected leaves (2–10 per species from 1 to 10 individuals, *n* = 53 samples; see complete sample list in Table 2) directly from the canopies of mature trees (including sunlit and shaded leaves) in French Guiana and Spain, and immediately froze them in liquid nitrogen (N) directly in the field to stop all biochemical activity. The environmental conditions during the samplings were within the range of typical conditions for each one of the sites. The leaves were then lyophilized and ground with a ball mill in the laboratory. The metabolites of the homogenized samples were extracted using a 1:1 methanol:water solution, and the LC–MS raw chemical spectra were processed and compared using XCMS [30]. Peaks were identified in each sample and then matched across samples to calculate the deviations of retention time and to compare relative ion intensities. MS1 spectra were filtered, establishing a noise threshold at 2 × 10^4^ and a minimum peak height at 2 × 10^5^. Chromatograms were aligned using an algorithm with a tolerance of 5 ppm of the mass-to-charge ratio (*m*/*z*) and a retention time of 0.2 min. Normalized areas under the peaks were used for quantification and were log-transformed before statistical analysis (see Appendix A for further details).

We obtained a total of 2461 metabolic compounds, which were further classified based on the stoichiometry of N and hydrogen:carbon and oxygen:carbon ratios [28,33] into four broad families of secondary metabolites: highly unsaturated polyphenols (92 compounds), polycyclic aromatics (42 compounds), aliphatics (864 compounds) and phenolics (80 compounds). It must be noted that this classifying procedure is useful when metabolites have a clear stoichiometric signature that allows grouping them into broad groups. Important compounds known to be related with plant defense such as terpenoids are mostly included within aliphatics (e.g., those without oxygen atoms such as isoprene, and most mono-, sesqui-, di- and triterpenoids), while some other complex terpene compounds having aromaticity may be included in other categories, such as unsaturated polyphenols. For each sample, we calculated its metabolite richness (i.e., its number of different compounds) and its chemical diversity (CD) as:$$\mathrm{CD}=-\overset{n}{\underset{i=0}{\sum }}pi\times \ln pi$$
where *p_i_* is the peak area of a compound relative to the total peak area, and the sum includes all compounds detected (after [25]).

### 2.2. Data Analysis

To test for differences in chemical richness and diversity for the entire metabolome and for each family of compounds, we built Bayesian multilevel models using the R package brms [34]. The richness and chemical diversity of the entire metabolome and for each family of metabolites calculated from each individual sample were modeled with region of origin (temperate vs. tropical) as a two-level, fixed-effects categorical factor, and the phylogenetic covariance matrix and the species identity as random factors, thus accounting for evolutionary relationships and the intraspecific variance [27,34,35]. We used the phylogenetic tree from Zanne et al. [31] to introduce the evolutionary relationships between the species in our models. We used a prior with a normal distribution with mean 0 and variance 10 times the standard deviation of the response variable to sample each coefficient, and a prior with a half Student-t distribution with 3 degrees of freedom for each of the random factors. We ran the models with 4 chains of 2000 iterations, burning-in the first 1000. All models converged with a potential scale-reduction statistic *R**^^^* < 1.01. Hypothesis testing of the fixed-effect parameters was carried out by checking 95% credible intervals and computing their associated evidence ratios. An evidence ratio of X can be interpreted as the alternative hypothesis (e.g., that tropical species have a greater diversity of phenolic compounds) being X times more probable than the null hypothesis. This was calculated using the hypothesis function from the R package brms [34]. Finally, we analyzed the similarity in chemical composition by means of a partial least squares discriminant analysis (PLS-DA) with the mixOmics R package [36], and tested for differences between region of origin and species by means of permutational multivariate analysis of variance (PERMANOVA) using Bray–Curtis distance matrices with the R package vegan [37].

## 3. Results

The metabolomic profiles of the species indicated that tropical trees tended to have more metabolites than did temperate trees (2280 ± 148 vs. 2111 ± 147, respectively; hereafter mean ± standard deviation of the posterior distribution estimate). This trend toward a greater richness of metabolites was modest but consistent when considering each family of biochemical compounds separately (794 ± 56 vs. 730 ± 53 for aliphatics, 40 ± 2 vs. 38 ± 2 for polycyclic aromatics, 84 ± 7 vs. 75 ± 7 for polyphenolics and 73 ± 6 vs. 67 ± 6 for phenolics for tropical vs. temperate species, respectively). This tropical effect on the number of metabolites was nevertheless weak or inconclusive, mainly due to the particularly rich metabolic profiles of the two temperate Fagaceae species (see the marginal-density plots on the X-axes in Figure 1).

In contrast to the number of metabolites, we found strong evidence that the diversity of phenolic compounds was higher in all four tropical species (the diversity increase in tropical species was 0.95 ± 0.44; (0.29–1.61) 95% CI, with an associated evidence ratio of 74.5; see the marginal-density plots on the *Y*-axis in Figure 1d). These compounds included procyanidins, flavone glycosides, chlorogenic acids and many other molecules with defensive properties against herbivores and pathogens [38]. Likewise, large unsaturated metabolites with many phenyl groups such as tannic acid, ellagitannins and other hydrolyzable tannins also tended to be more diverse in tropical species, although this difference was lower than with phenolic compounds (the diversity increase in tropical species for polyphenolics was 0.47 ± 0.49; (–0.29 to 1.23) 95% CI, with a moderate evidence ratio of 7.2; Figure 1c). In contrast to phenolics and polyphenolics, there were no differences in phytochemical diversity between tropical and temperate tree species either for the entire metabolome or for aliphatic and aromatic compounds (Figure 1a,b).

The analysis of the complete foliar metabolomes by means of a PLS-DA showed that the tropical and temperate species clustered together, thus indicating a distinctive chemical composition of their leaves (Figure 2a). The clearly greater spread of the samples of tropical species suggests that for the whole foliar metabolome, both the interspecific and intraspecific variation seemed greater between and within the tropical species than in the temperate ones. An equivalent ordination using only the subset of phenolic compounds also showed that the tropical and temperate tree species had distinctive phenolic profiles, but interestingly, the tropical species apparently had less (or at least similar) intraspecific variation of their phenolic profiles than the temperate species (Figure 2b). The region of origin of the tree species was significant (*p* < 0.001) in both PERMANOVA tests, explaining 11% and 13% of the variation in the composition of the complete and of the phenolic component of the foliar metabolome, respectively. The differences between species, irrespective of their region of origin, were also significant (*p* < 0.001) and accounted for 49% and 53% of the variation in the composition of the complete and of the phenolic component of the foliar metabolome, respectively.

## 4. Discussion

This report provides strong evidence supporting that the diversity of phenolic compounds, which are a key class of constitutive and induced plant defenses against pathogens and herbivores [9], was higher for the tropical tree species studied, irrespective of their evolutionary history. The functionally closely related polyphenolics also provided moderate evidence supporting the same pattern, but importantly, this “tropical effect” was not detected for the entire metabolomic profile or for any other family of biochemical compounds assessed in this study. The fact that this tropical–temperate difference in phytochemical diversity is mostly due to the phenolic part of the foliar metabolome, and to a lower extent by the polyphenolic compounds, indicates that this metabolic contrast must be linked to the functional role that these compounds play. Since it is well established that the primary function of these compounds is protection against abiotic and biotic pressures, it seems reasonable to presume that tropical tree species must have been selected to keep a more diverse array of phenolic diversity than their temperate counterparts, likely due to stronger abiotic and biotic pressures [2]. In fact, numerous studies suggest that tropical trees must cope with stronger biotic interactions and a more diverse array of either generalist or specialized enemies [11,14,17,39,40].

This higher diversity in phenolic compounds in the leaves of tropical trees was neither the result of having more metabolites, which actually is hardly the case in the species studied here, nor the outcome of a divergent evolutionary history between the tropical and temperate subsets of species. The control for the phylogenetic relationships among the species included in the analysis (see Table 1) allowed us to exclude the emergence of the observed pattern only from differences in the timing and mode of diversification between the two latitudinal regions [7]. Additionally, the higher diversity of phenolic compounds in the leaves of the tropical species was coupled with a distinctive composition. Richards et al. [41] assessed the metabolomic profile, herbivore community and herbivory pressure of 22 *Piper* species from Costa Rica and found that phytochemical diversity reduced herbivore damage, while it could also promote herbivore specialization. Similarly, Salazar et al. [38] reported that phenolic compounds were the most important chemical defenses in 31 sympatric species within the monophyletic Protieae clade in Peru, and that their high diversity was due to cumulative diffuse interactions with a broad range of generalist herbivores. The only other latitudinal comparative study with such a broad metabolomics approach reported greater chemical divergence among tropical species, suggesting a stronger biotic selective force in the tropics and therefore more rapidly evolving defenses [42]. Our results are therefore in line with these previous studies, and also with the only assessment with a geographically and methodologically comparable approach [42]. The limited scope of this study, however, precludes a proper test of the mechanism behind the observed greater phytochemical diversity in phenolic and polyphenolic compounds.

The evolution of plant defenses can escalate (i.e., the accumulation of chemical traits during the macroevolution of a plant lineage) and diverge (i.e., the increase in chemical dissimilarity between closely related plant species), both by generalist and specialized biotic antagonists, respectively [22,29]. Previous research has shown that that interspecific variation is greater than intraspecific variation among tropical tree species, even suggesting the existence of species-specific “metabolomic niches” [42,43,44]. This study, however, was not conceived to assess intraspecific variation in phytochemical diversity between tropical and temperate tree species. Thus, the limitations of our dataset in terms of number of species, samples per species, and particularly in the number of individuals per species, severely hinders the ability to draw firm conclusions on the potential role of intraspecific and interspecific variation on leaf chemical composition and the evolution of plant defenses. Even so, our finding that for the whole foliar metabolome, both the interspecific and intraspecific variation seems greater in the tropical species than in the temperate ones, agrees with what we would expect by their generally greater chemical richness. Moreover, when analyzing only the phenolic profiles, we found that tropical tree species apparently showed less (or similar) intraspecific variation than temperate ones, and this likely could be due to a lower capacity to exhibit inducible variation in defenses [45]. These suggestions should be properly investigated in further research efforts with more balanced sampling designs. It is also important to acknowledge that endophytic fungi may also synthesize defense compounds that cannot be easily distinguished from those produced by plants. Hence, part of the greater phytochemical diversity observed in the tropical tree species analyzed could also result from a greater contribution by mutualistic fungi. Nevertheless, future research with a similar metabolomic approach applied to a phylogenetically broader dataset and including more sources of variation of intraspecific chemical composition [43,44,45] should allow assessing phytochemical diversity across multiple plant lineages with contrasting patterns of diversification. This could help to disentangle the mechanisms driving the evolution of plant chemical defenses and the role that they may have in plant-species coexistence.

## Figures and Tables

**Figure 1 plants-10-00554-f001:**
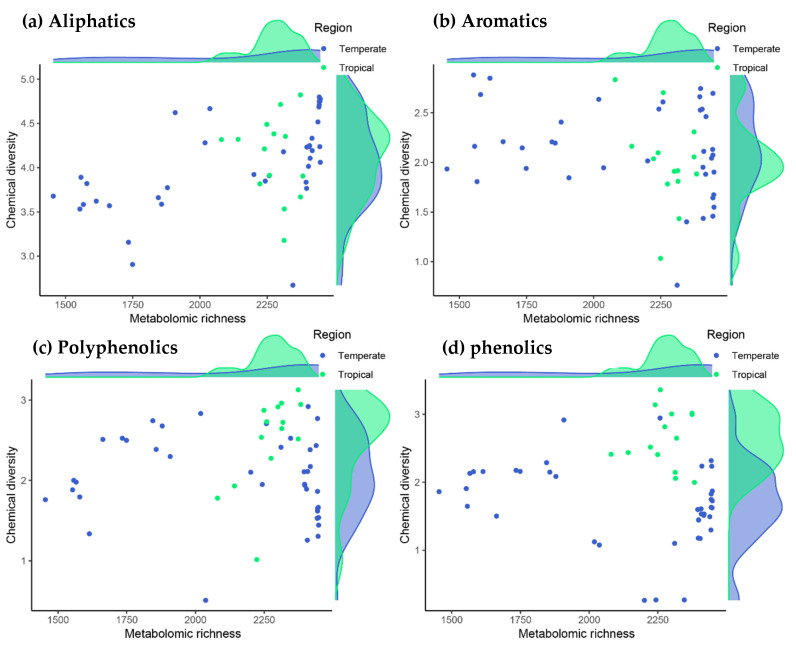
Relationships between the number of metabolites (metabolomic richness) with metabolite diversity (chemical-diversity index) for the main families of chemical compounds in the tropical and temperate tree species studied. Phenolic and polyphenolic compounds were more diverse in tropical species, with an evidence ratio of 74.5 and 7.2, respectively, based on Bayesian multilevel phylogenetic models (note the tropical–temperate differences in the marginal-density plots of the *Y*-axis in panels c and d, and see text for further details).

**Figure 2 plants-10-00554-f002:**
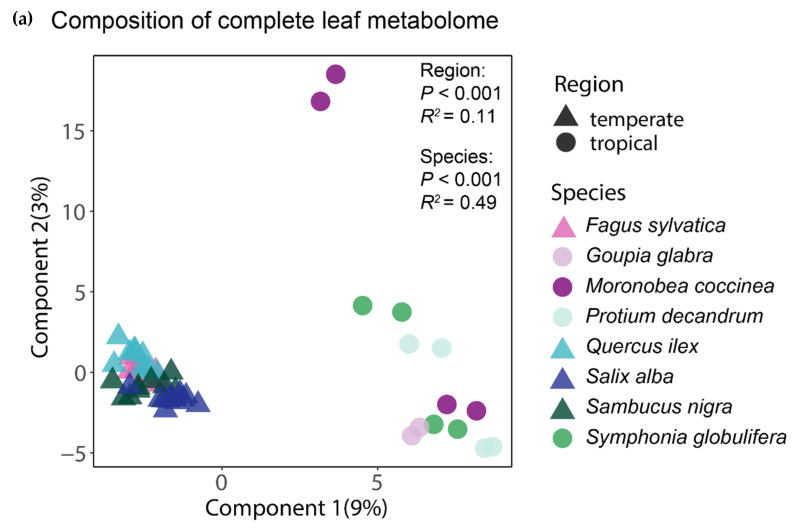
Similarity in chemical composition for the complete foliar metabolome (**a**) and for phenolics (**b**) between the tropical and temperate tree species studied. The ordination of individual samples resulted from the partial least squares discriminant analyses (PLS-DAs) with species and region as group classifiers. The statistical significance and explained variance of the insets resulted from a permutational multivariate analysis of variance using Bray–Curtis distance matrices.

**Table 1 plants-10-00554-t001:** Phylogenetic relationships of the tree species.

Region	Species Richness	Phylodiversity	Mean Pairwise Distance	Mean Neighbor Taxon Distance
Tropical	4	377	200	146
Temperate	4	392	208	157

Phylogenetic metrics (in millions of years) are derived from the plant phylogeny published by [31] and updated by [32]. Phylodiversity refers to the sum of the lengths of all branches on the phylogeny that span the members of the species set. Temperate species are *Sambucus nigra* (Adoxaceae), *Salix alba* (Salicaceae), *Quercus ilex* and *Fagus sylvatica* (Fagaceae). Tropical species are *Goupia glabra* (Goupiaceae), *Protium decandrum* (Burseraceae), *Symphonia globulifera* and *Moronobea coccinea* (Clusiaceae). See Appendix A.

**Table 2 plants-10-00554-t002:** List of samples analyzed.

Species Names and Family	Sampled Leaves	Individuals
*Fagus sylvatica* L. (Fagaceae)	9	9
*Goupia glabra* Aubl. (Goupiaceae)	2	1
*Moronobea coccinea* Aubl. (Clusiaceae)	4	1
*Protium decandrum* Aubl. (Burseraceae)	4	1
*Quercus ilex* L. (Fagaceae)	10	10
*Salix alba* L. (Salicaceae)	10	4
*Sambucus nigra* L. (Adoxaceae)	10	10
*Symphonia globulifera* L.f. (Clusiaceae)	4	1

Temperate species were sampled from different sites in Catalonia (Spain), while tropical species were sampled from the Paracou and Nouragues research stations in French Guiana (France).

## Data Availability

Raw data supporting the results of this study are available at Figshare repository https://doi.org/10.6084/m9.figshare.12240581.v1. Last accessed date 15 March 2021.

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
