# Peer review of "Metabolome-Wide, Phylogenetically Controlled Comparison Indicates Higher Phenolic Diversity in Tropical Tree Species"

_plants, 2021, doi:10.3390/plants10030554_

Round 1
Reviewer 1 Report
The article constitutes an interesting approach to molecular ecology by using stoichiometrically based classification of phenolics compounds.
Some specific aspects must be answered to improve the article:
- According to Table S1: For tropical trees, only one individual was sampled for each specie (and just 2-4 leaves/tree) in contrast, for temperate trees (4-10 individuals sampled). Given the intraspecific chemical variability ¿inadequate sampling procedure could bias the conclusions?
- Although it is not directly related to the study, has the presence of endophytic microorganisms in the sampled species been ruled out or analyzed?. These microorganisms can synthesize defense compounds in plants.
- Why only phenolic compounds were considered for the analysis? For example, terpenoids plays an important role in plant-pathogen or plant-herbivores interactions.
Author Response
The article constitutes an interesting approach to molecular ecology by using stoichiometrically based classification of phenolics compounds.
RESPONSE: Many thanks for your positive evaluation of the interest of our article.
Some specific aspects must be answered to improve the article:
- According to Table S1: For tropical trees, only one individual was sampled for each specie (and just 2-4 leaves/tree) in contrast, for temperate trees (4-10 individuals sampled). Given the intraspecific chemical variability ¿inadequate sampling procedure could bias the conclusions?
RESPONSE: Yes, the limitations of our dataset in terms of number of samples per species limits the conclusions we can draw from the study regarding the role of intraspecific chemical variability. We acknowledge that in the last paragraph of the discussion. However, the novelty and the clarity of the pattern we have found when comparing tropical versus temperate species (i.e. the interspecific comparison that is the main point of the contribution) is in our opinion worthy of being published.
- Although it is not directly related to the study, has the presence of endophytic microorganisms in the sampled species been ruled out or analyzed?. These microorganisms can synthesize defense compounds in plants.
RESPONSE: This is an interesting point we have not explored in this work. We concur with the reviewer that is important but we cannot distinguish the origin of the compounds to tell apart which are from mutualistic fungi from those of plant origin. In any case, we could conclude that part of the greater phytochemical diversity observed in the tropical tree species analyzed could also result from a greater contribution by mutualistic fungi. That’s an interesting point that we have now included in the last paragraph of the discussion. Thank you.
- Why only phenolic compounds were considered for the analysis? For example, terpenoids plays an important role in plant-pathogen or plant-herbivores interactions.
RESPONSE: Yes, we agree with the reviewer that terpenoids are important components but their atomic structure do not provide a clear stoichiometric signature allowing us to cluster them as “family”. It is likely, however, that those without oxygen atoms (e.g. isoprene, mono- sesqui- di- and triterpenoids etc.) have been included in the general aliphatic compound family.
Reviewer 2 Report
The manuscript describes good and substantial information regarding the hypothesis that the phenolic diversity of tropical tree leaves is greater than the phenolic diversity of temperate tree leaves. In addition, the significant quality of the article is the material and method section, which is clearly written and easy to reproduce. The result and discussion section is well written and the obtained results are efficiently compared with previous studies.
Overall, the manuscript is well written with significant data which is sufficiently described and discussed.
For further improvement, the following corrections are required:
- Abstract: Space after: and Tropical should not be in bold
- In the result section, or in the supplementary materials the authors could introduce the tables with the complete metabolomics profile of tropical and temperate species
- the conclusion section is missing
- in references from reference 10, the reference number appears twice
In the attached supplementary materials there are some general corrections.
Overall, the manuscript is of interest and presents detailed work with important results.

Author Response
The manuscript describes good and substantial information regarding the hypothesis that the phenolic diversity of tropical tree leaves is greater than the phenolic diversity of temperate tree leaves. In addition, the significant quality of the article is the material and method section, which is clearly written and easy to reproduce. The result and discussion section is well written and the obtained results are efficiently compared with previous studies.
Overall, the manuscript is well written with significant data which is sufficiently described and discussed.
RESPONSE: Many thanks for your positive evaluation of the manuscript.
For further improvement, the following corrections are required:
- Abstract: Space after: and Tropical should not be in bold
RESPONSE: Done.
- In the result section, or in the supplementary materials the authors could introduce the tables with the complete metabolomics profile of tropical and temperate species
RESPONSE: The dataset with the full metabolomics profiles contain thousands of columns. It is for this reason that we refer readership to the repository where these data is fully and publicly available under a CC by 4.0 license: https://doi.org/10.6084/m9.figshare.12240581.v1
- the conclusion section is missing
RESPONSE: For the sake of concision we have included our conclusions at the beginning and at the end of the discussion. We are afraid that including a specific subsection with the conclusions we could end up being a bit redundant.
- in references from reference 10, the reference number appears twice
RESPONSE: In our version the manuscript is already partially formatted by the production officers and we actually have all the numbers of the reference list duplicated. Since the manuscript is partially formatted we cannot modify it without losing the format. Hence, we would like to ask to the production office if they can solve this issue. Thanks in advance.
In the attached supplementary materials there are some general corrections.
RESPONSE: Thanks, we have accepted all the edits.
Overall, the manuscript is of interest and presents detailed work with important results.
RESPONSE: Thanks again for your help improving the manuscript
Reviewer 3 Report
In the title: "tropical tree species" please add family? region? Authors compared these with temperate trees?
The abstract part doesn't contain any numerical findings, and no information abuout what polyphenols were analyzed. "four majot families" this is very general. The abstract should contain short description of method and some results, and brief conclusion.
The Table S1 should be present in the main text.
Moreover, the results and discussion parts are very short (in total 2 pages). They contain only general overview about the metabolome profile in different tree species, whereas the phenolic compounds is a wide group of varied chemical molecules. Furthermore, in the results, it is only discussion about 'tropical' and 'temperate' trees, but Authors showed names of specific trees analyzed, to compare some data from other studies should be added.
In this case, I doesn't recommend t publish these article, it needs a great work in writing a nice manuscript.
Author Response
In the title: "tropical tree species" please add family? region? Authors compared these with temperate trees? The abstract part doesn't contain any numerical findings, and no information about what polyphenols were analyzed. "four majot families" this is very general. The abstract should contain short description of method and some results, and brief conclusion. The Table S1 should be present in the main text. Moreover, the results and discussion parts are very short (in total 2 pages). They contain only general overview about the metabolome profile in different tree species, whereas the phenolic compounds is a wide group of varied chemical molecules. Furthermore, in the results, it is only discussion about 'tropical' and 'temperate' trees, but Authors showed names of specific trees analyzed, to compare some data from other studies should be added. In this case, I doesn't recommend t publish these article, it needs a great work in writing a nice manuscript.
RESPONSE: Unfortunately, we cannot agree with the reviewer. The present study addresses a single but very important question that is the role of defensive compounds in relation to the latitudinal gradient of plant diversity. It tests this hypothesis by means of simple but phylogenetically controlled comparison between 4 temperate and 4 tropical tree species and we present a novel and consistent finding that is the tropical species show a greater diversity of phenolic compounds irrespective of their evolutionary history. As explained before, we acknowledge the limitations of the study, so we think that it is better to avoid speculations. We are afraid that going into the detail of which species or which compounds requested by the referee would dissipate the focus away from what we consider the main finding and the one for which we have a solid result to communicate.
Round 2
Reviewer 3 Report
Thank you for your response.
Author Response
Many thanks for your help improving our manuscript and for your acceptance recommendation.